# Retrospective Cohort Study of Intrapericardial Cisplatin for Risk Reduction of Malignant Pericardial Effusion Recurrence

**DOI:** 10.3390/curroncol32100568

**Published:** 2025-10-11

**Authors:** Francisco Javier Muñoz-Carrillo, Roxana Maribel Reyes, David Pesántez, Gemma Carrera, Enric Cascos, Pedro Castro, Sara Fernández-Méndez, Carme Font, Laura González-Aguado, Ignacio Grafiá, Lucía Llavata, Inés Monge-Escartín, Joan Padrosa, Noemí Reguart, Adrián Téllez, Albert Tuca, Margarita Viladot, Carles Zamora-Martínez, Patrícia Amorós-Reboredo, Javier Marco-Hernández

**Affiliations:** 1Medical Oncology Department Hospital Clinic de Barcelona and Translational Genomics and Targeted Therapies in Solid Tumors, IDIBAPS, 08036 Barcelona, Spain; frmunoz@clinic.cat (F.J.M.-C.);; 2Cardiology Department, Hospital Clínic de Barcelona, 08036 Barcelona, Spain; 3Medical Intensive Care Unit, Internal Medicine Department, Hospital Clínic de Barcelona, 08036 Barcelona, Spain; pcastro@clinic.cat (P.C.);; 4Hospital Pharmacy Department, Hospital Clínic de Barcelona, 08036 Barcelona, Spain; 5Chair of Palliative Care, Barcelona University, 08036 Barcelona, Spain; 6EAPS Fundació La Caixa, 08019 Barcelona, Spain; 7Pharmacy Department and Digital Impulse, Strategy and Transformation Area, Hospital de Sant Pau, Institut de Recerca Sant Pau (IR SANT PAU), 08041 Barcelona, Spain

**Keywords:** malignant pericardial effusion, cisplatin, intrapericardial infusion, multidisciplinary approach

## Abstract

**Simple Summary:**

Malignant pericardial effusion is a life-threatening complication in patients with cancer and frequently recurs after standard drainage procedures. Our study evaluated the use of intrapericardial cisplatin, a chemotherapy drug administered directly into the pericardial space, as an approach to prevent recurrences. We analyzed data from 41 patients with solid neoplasms (mainly lung cancer), treated with a standardized, multidisciplinary protocol at a tertiary hospital over a 13 year period. Patients with hematological malignancies were excluded. Our goal was to assess the safety, feasibility, and effectiveness of this technique. The treatment was generally well tolerated, with a very low rate of fluid recurrence and manageable side effects. These findings support the integration of intrapericardial cisplatin into multidisciplinary care pathways, making it a safe and effective procedure to consider for managing malignant pericardial effusion in patients with solid tumours.

**Abstract:**

Malignant pericardial effusion (MPE) is a life-threatening condition in patients with cancer, with common recurrences after simple pericardiocentesis. Consequently, the intrapericardial instillation of sclerosing or cytotoxic agents has been explored, with limited evidence from small studies with different methodologies. We undertook an observational, retrospective, single-centre study, including all patients diagnosed with a solid neoplasm and clinically significant and/or recurrent, cytology-confirmed MPE, treated with Intrapericardial Instillation of Cisplatin (IPIC), between 2009 and 2022. Patients with hematological malignancies were excluded. The procedure followed a multidisciplinary approach and a standardized protocol. Variables collected included baseline patient characteristics, neoplasm details, MPE impact, adverse events (AEs) from procedures (pericardiocentesis and IPIC) and outcomes (time to MPE recurrence and survival). This study adhered to the STROBE guidelines. A total of 41 patients were included, 51% female, with a median age of 61 (51–69) years. Non-small cell lung cancer (NSCLC) was the predominant primary tumour (78%) and in 44% of the cohort, MPE was identified at cancer diagnosis. Most patients (90.2%) presented symptoms related to MPE at diagnosis, and 88% had cardiac tamponade on echocardiography. IPIC was administered a median of four times. IPIC-related AEs occurred in 10 patients (24.4%), with transient atrial fibrillation (AF) being the most frequent one. Two patients (4.9%) experienced MPE recurrence within 30 days after IPIC. The median survival time from MPE diagnosis was 161 days (5.4 months; IQR 73–455 days). IPIC appears to be a feasible, effective and safe option for reducing the risk of MPE recurrence, mainly in NSCLC.

## 1. Introduction

Malignant pericardial effusion (MPE) develops when cancer cells invade the pericardium, causing abnormal fluid accumulation in the pericardial sac [1]. It is common in patients with cancer, with a prevalence of 5–20% [2], and in 10% of cases, constitutes the first manifestation of an occult neoplasm [3]. MPE is primarily caused by direct or metastatic spread of a distant tumour, with lung cancer (∼35%) [4,5], breast cancer (∼20%) [5,6], and hematological malignancies (∼15%) [4,7,8] being the most frequently involved, whereas primary tumours of the pericardium are rare [9,10,11].

MPE clinical significance is heterogeneous, ranging from an asymptomatic mild effusion to a severe cardiac tamponade (CT) with haemodynamic instability, a life-threatening condition that requires emergent treatment [7,12,13]. When CT develops, drainage of pericardial effusion, by means of a pericardiocentesis, is often mandatory and usually effective for immediate symptom control [7]. Nevertheless, most patients (70%) present early recurrences [14,15].

Several strategies have been developed and gathered in the 2015 European Society of Cardiology (ESC) guidelines for the diagnosis and management of pericardial diseases to mitigate the risk of local recurrence of MPE [7], including pericardiocentesis with catheter drainage, a pleuropericardial window through balloon pericardiotomy or surgery, pericardiectomy [15,16], or intrapericardial instillation of sclerosing or cytotoxic agents [4,9,17,18]. However, there is a lack of consensus on the optimal approach.

Regarding intrapericardial instillation of cytotoxic agents, the most widely used are thiotepa [4,14,19], bleomycin [17], and platins. Carboplatin [20], but especially cisplatin, a widely used chemotherapy agent, is used off label for intrapericardial instillation (IPIC).

However, this has been performed with limited evidence and in adherence to ethical principles for a favourable risk-benefit balance [21,22,23,24]. Therefore, there is a need for more evidence to support and ensure the safety of this procedure, especially given advancements in cancer treatment in recent years. Limited evidence supports its use, with studies often being small and methodologically diverse, highlighting the need for more robust data.

The aim of the present study is to evaluate the feasibility, effectiveness, and safety of our local protocol, intrapericardial cisplatin (IPIC), in reducing the risk of MPE recurrence. We hope to provide valuable insights into the management of MPE and contribute to the development of standardized treatment protocols, ultimately improving patient outcomes.

## 2. Materials and Methods

### 2.1. Study Design and Participants

We conducted a retrospective cohort study including all patients over 18 years old, both male and female, since the treatment was expected to have similar effects across genders, diagnosed with a solid neoplasm and MPE, treated with IPIC (at least one dose) in a tertiary care hospital between 2009 and 2022. Patients with hematological malignancies were not included in the analysis.

Data were collected from medical records, according to our hospital’s pharmacy department register. Since IPIC represents an off-label indication, all patients were properly informed about the procedure, benefits, and potential AEs, providing consent for its administration. The study was reviewed and approved by the local Clinical Research Ethics Committee (reference HCB/2022/0304) and adhered to the STROBE (Strengthening the Reporting of Observational studies in Epidemiology) Statement (Appendix A, Table A1).

The IPIC procedure was standardized by developing a local protocol through a multidisciplinary approach involving oncologists, internal medicine specialists, cardiologists, intensivists, nurses, and clinical pharmacists. The eligibility criteria included patients affected by a clinically significant and/or recurrent MPE requiring intrapericardial drainage placement (through pericardiocentesis or surgical pleuropericardial window), confirmed malignancy in pericardial effusion cytology, and an Eastern Cooperative Oncology Group (ECOG) performance status (PS) of 0–2 prior to hospital admission. The only formal contraindication was decompensated cardiac arrhythmia. The protocol also outlines the technique, preparation, and treatment administration, regardless of patient location (conventional hospitalization ward or intensive care unit (ICU)), to coordinate the entire team. The procedure consisted of eight steps: 1. Verification that the catheter was correctly placed in the pericardial cavity (checked through transthoracic echocardiography (TTE)) and had a pericardial output <150 mL in 24 h. 2. Manual drain of the residual pericardial effusion prior to cisplatin administration. 3. Pharmaceutical preparation of cisplatin solution: 10 mg of cisplatin diluted in 20 mL of saline. 4. Manual administration of the dilution through the intrapericardial catheter over 5 min while monitoring heart rate, due to the risk of atrial fibrillation (AF). If AF developed, the administration was suspended, and the AF was treated accordingly. 5. Closure of the intrapericardial catheter for 24 h. 6. The procedure was repeated daily for a maximum of 5 days, or until the daily drainage was <25 mL, with a maximum cumulative dose of cisplatin ≤50 mg. 7. After the drainage was not productive and a TTE confirmed a low amount of pericardial effusion, without hemodynamic compromise, the pericardial catheter was withdrawn. 8. TTE was repeated after 30 days to detect recurrences.

### 2.2. Variables

Gathered data included: baseline patient characteristics (age, sex, comorbidities, concomitant medications), characteristics of the neoplasm (primary tumour, histology, molecular data) and its previous treatments, variables related to MPE (amount of effusion, CT presence, main symptom, method of drainage, length of catheter placement, need of ICU admission, length of hospital stay), variables related to the IPIC treatment (days of cisplatin infusion, suspected adverse events (AEs), concomitant systemic treatment indicated), and outcomes (recurrence of MPE, amount of effusion at recurrence, time to recurrence, death and survival time since MPE drainage). AEs were graded according to the Common Terminology Criteria for Adverse Events (CTCAE), v5.0.

### 2.3. Assessing Clinical Outcomes

Safety assessments included any AEs described in the electronic medical report during IPIC treatment. AEs were defined according to Good Clinical Practice as any sign or symptom that was not present before IPIC, and with a high suspicion of being related to it. They were categorized based on the IPIC initiation date, as follows: immediate (within the first 5 days—when cisplatin instillation is usually performed), early (between the 6th and 14th day) or late (beyond 2 weeks after instillation). Effectiveness outcomes included the recurrence of pericardial effusion and the survival time since MPE drainage.

### 2.4. Statistical Analysis

Categorical variables were described using absolute numbers and percentages, while quantitative parameters were described with medians and interquartile ranges (IQR). Overall survival was defined as the amount of time (in days) between the MPE diagnosis and the date of death or last follow-up, if alive, and it was described using the Kaplan–Meier method. Survival curves were compared using the log-rank test. To assess the impact of targetable oncogenic alterations, subgroup analyses were performed specifically for patients with EGFR, ALK, and BRAF mutations. Kaplan–Meier survival curves were generated for these subgroups to compare overall survival (OS) with the general cohort.

All analyses were performed using SPSS v25 (SPSS Inc., Chicago, IL, USA) and Jamovi v2.5 (Sydney, Australia). A *p*-value < 0.05 was considered significant.

### 2.5. Bias Reduction Measures

All IPIC-treated patients were recorded in the pharmacy department information system, eliminating selection bias. Inclusion criteria ensured all IPIC recipients were accounted for. Additionally, diagnostic, intrapericardial treatment, and follow-up data were systematically documented in patient clinical records. An analysis of data loss during follow-up was conducted, reducing information bias.

## 3. Results

### 3.1. Description of the Cohort Patients

Forty-one patients were included, which represented all the patients who underwent IPIC between 2009 and 2022. Twenty-one (51%) were women, with a median age of 61 (IQR: 51–69) years. Twenty-five of them (61%) had at least one comorbidity, with arterial hypertension (18 patients, 43.9%) being the most frequent one. The most common primary tumour was non-small cell lung cancer (NSCLC, 32 patients; 78%). Most patients were metastatic at MPE diagnosis (35 patients; 85.4%), and in almost half of them, the MPE was identified at the time of the first cancer diagnosis (18 patients; 43.9%). Thirty-four patients (83%) were taking at least one concomitant medication, with proton-pump inhibitors, paracetamol and statins being the most frequent ones. At the time of the MPE diagnosis, seventeen patients (41%) were still not under oncologic treatment: ten (24%) were receiving chemotherapy, nine (22%) were receiving targeted therapies, four (10%) were receiving immunotherapy and one (2%) was receiving hormone therapy. The most notable characteristics of the patients are summarized in Table 1.

### 3.2. Somatic Alterations in NSCLC

Among NSCLC patients, oncogenic-driven alterations were detected in 15 (47%), of which 12 (29%) received targeted therapies. Epidermal growth factor receptor (EGFR) exon 19 deletion was the most frequently detected alteration (6 cases; 19%) followed by anaplastic lymphoma kinase (ALK) translocation and Kristen rat sarcoma virus (KRAS) mutation (3 cases each; 9%), other EGFR mutations (2 cases; 6%), and B-Raf proto-oncogene (BRAF) mutation (1 case; 3%).

### 3.3. MPE and Pericardiocentesis Characteristics

Most patients presented symptoms related to MPE at the time of diagnosis (37 patients; 90.2%), with almost all presenting clinical or echocardiographic CT (36; 87.8%). ICU admission was necessary in sixteen cases (39%), with a median length of stay of 4.5 days (IQR 2.75–6.25). The main reason was haemodynamic instability, secondary to CT. All patients were initially treated by pericardiocentesis, most of them through a subxiphoid approach (34; 82.9%); in the remaining cases, the access method was not specified. Seven patients (17.1%) presented procedure-related complications, with self-limited, grade 1 (G1) AF being the most common (five out of seven cases). None of the patients who developed AF had a previous history of arrhythmias, and sinus rhythm was restored within five days after the procedure. Further data regarding MPE characteristics are presented in Table 2.

### 3.4. Intrapericardial Cisplatin Instillation—Safety

After pericardiocentesis, IPIC was performed a median of four times (range one to five). Reasons for interrupting treatment before completing the five planned instillations included arrhythmias (AF), accidental catheter withdrawal or misfunctioning, and excessive pericardial fluid drainage (when that happened, IPIC could be resumed as soon as the drainage decreased, although it was not completed, due to catheter dysfunction). Ten patients (24.4%) experienced G1 AEs related to IPIC, most of them (70%) during the days in which instillations were being performed (AF (n = 3), chest pain (n = 3), sinus tachycardia (n = 1), and nausea (n = 1)). No AEs were identified later than two weeks after the first IPIC. No fatal or life-threatening events occurred. Details on IPIC and its associated AEs are depicted in Table 2.

### 3.5. Recurrence and Survival Outcomes of the Cohort

The median follow-up from IPIC administration was 5.4 months. MPE recurrence occurred in only two patients (4.9%), both within the first 30 days after IPIC. These recurrences were asymptomatic and diagnosed through echocardiography or computed tomography. Neither case required another pericardiocentesis, due to the scarce volume of effusion. Both patients died of a progressive disease (not related to MPE), 25 days and 52 days after the initial MPE diagnosis, respectively; in one of them, only one dose of IPIC was administered, due to catheter dysfunction.

In most cases (70.7%), systemic therapy was initiated or modified (if confirmed progression to the prior treatment) after the MPE episode.

Regarding the descriptive analysis, median survival time from MPE diagnosis was 161 days (5.4 months; IQR 73–455 days). Five patients (12.2%) died within the first month, and thirteen (31.7%) within the first three months. Nineteen (46.3%) survived beyond six months, and fourteen (34.1%) beyond one year. Survival is illustrated in the Kaplan–Meier curve shown in Figure 1. In a subgroup analysis of patients with targetable oncogenic alterations (EGFR, ALK, BRAF), the median overall survival exhibited a non-significant trend toward being longer, compared to the rest of the patients. Kaplan–Meier curves for these subgroups are shown in Figure 2, indicating a median survival of 14 months (95% confidence interval, 10.2—not reached; *p*-value of 0.07). Comparison of NSCLC with or without targetable oncogenic alterations did not reach statistical significance either (*p*-value of 0.18).

## 4. Discussion

This study analyzed forty-one patients who underwent IPIC therapy for MPE, showing that the overall safety profile was favourable, with a low MPE recurrence rate (4.9%) during follow-up, and a median survival time from MPE diagnosis of 161 days. These findings suggest that IPIC represents a viable and effective therapeutic strategy in managing this unfavourable condition.

MPE is a common (5–20%) condition in patients with cancer [2] that can constitute a life-threatening condition that needs emergent treatment when CT develops [7,12,13]. Then, drainage of pericardial effusion is mandatory and usually effective for immediate control of symptoms [7], but up to 70% of patients present early recurrences [14,15]. The best approach to decrease the risk of local recurrence of MPE is not known, but intrapericardial instillation of cytotoxic agents is common, as noted in the 2015 ESC guidelines [4,7,9,17,18]. It seems to reduce MPE recurrences and induces a local antitumor effect, limiting systemic side effects by increasing the intrapericardial concentration. Furthermore, cardiac movement facilitates agent diffusion [23,25,26]. The most used cytotoxic in this scenario is cisplatin, an inorganic substance that is not cell-cycle specific, which inhibits DNA synthesis by producing cross-links within and between DNA strands; it also inhibits protein and RNA synthesis and enhances tumour immunogenicity [21]. Some studies suggest that IPIC is more effective in MPE, and secondary to lung cancer and thiotepa in breast cancer [2,25,27]. IPIC seems to avoid recurrence in up to 93% and 83% of patients with mild related AEs at three and six months, respectively, with AF being the most frequent [22,24]. Sclerosing agents like tetracyclines/bleomycin achieve local control rates of up to 80% [28], but have higher rates of AEs: fever (27.8%), chest pain (20%), and arrhythmias (16.7%) [28,29].

The present study describes one of the largest published series of intrapericardial chemotherapy, with the exception of a very recent work focused on NSCLC [30]. The number of included patients in our study matches previous studies [22]. The study recruited patients with MPE and identified clinical complexities within the treated population, indicating a higher likelihood of aggressive disease onset. Comorbidities in patients with cancer, particularly active smoking, arterial hypertension, and dyslipidaemia, were prevalent (61%). Since we included patients eligible for pericardiocentesis as the initial therapeutic approach, e.g., presenting with severe, symptomatic MPE, this could explain the higher clinical compromise at MPE diagnosis compared to other studies.

Consistent with other studies [4,5,6,7], within our cohort, lung and breast cancer emerged as the most common primary etiologies of MPE. Symptomatic and clinically evident MPE is most commonly associated with lung cancer, underscoring the higher aggressiveness of this disease [3,22,23]. Moreover, in our cohort, MPE was the first presenting sign of lung cancer in eighteen patients (43.9%), and in four additional cases (9.8%) MPE appeared in the first three months after cancer diagnosis.

Interestingly, nearly half (47%) of NSCLC cases exhibited mutations in oncogenic drivers that can be therapeutically targeted; in 66.7% of the cases, MPE developed as a progression under targeted therapy. Indeed, ALK gene rearrangements have more metastatic pericardial and pleural spread compared to other mutations [31]. This is noteworthy, as pericardial involvement can be managed as part of oligoprogression of the tumour: treating MPE with local therapies allows for the continuation of previous targeted treatment, thereby preserving the favourable prognosis that these tumours inherently have compared to their driver-negative counterparts [32].

When administering IPIC, we adopted the same regimen as Tomkowski et al. in 39 of the 46 patients they reported [23]. In contrast, Bischionitis et al. applied IPIC for three days rather than five, using the same cisplatin dilution [24], while Maisch et al. employed a single-dose cisplatin approach [22]. Our recurrence and AE rates are consistent with prior IPIC series [22,23,24].

In our series, AEs related to pericardiocentesis or drug instillation occurred in 24.4% of patients: all G1 and none being severe. This is consistent with the studies published to date, where the rate of AEs related to IPIC is 13.6%. The most frequent is AF, with incidences of 12–32% [33], as described in Bischionitis et al. [24] (3/25 patients), Lafaras et al. [34] (5/56 patients), and Tomkowski et al. [23] (7/46 patients).

Other intrapericardial cytotoxics seem to be associated with worse rates or severity of AEs, like tetracyclines, with up to 45% of cases: mostly fever and pain [35,36]. Other treatments have been used less, such as thiotepa, with a rate of AE of 6.7% [18]; bleomycin, with reported cases of AF (16.7%), fever (27.8%) [37], and one case of constrictive pericarditis and death [38]; or mitomycin C, with another case of constrictive pericarditis [39]. Other agents like mitoxantrone, minocycline, carboplatin, or aclarubicin have also been reported in fewer than 10 patients, with lacking safety profiles [20,35,40,41].

Regarding outcomes, only 2 patients out of 41 presented significant MPE recurrence, which supposes a 95% success rate. This is consistent with previous studies conducted with IPIC, with a rate of controlled MPE at about 90% (range: 82–100%) [22,23,24,34,42]. Other intrapericardial agents have shown similar success rates, like tetracycline (83–91%) [35,36], thiotepa (83–100%) [14,18,43], bleomycin (78–100%) [30,37,38], mitomycin C (60–88%) [39,44], and mitoxantrone (100%) [40,41].

This therapeutic approach to MPE allowed us to initiate oncologic treatment in over 70% of our patients, which could improve the survival of these patients, especially in the oncogenic-driven group.

Consequently, cytotoxic agents such as cisplatin may offer a more interesting profile compared to other drugs, such as sclerosing agents, with less AEs and similar rates of success [17,28,29].

Real-world evidence allows for the inclusion of patients who are often excluded from randomized trials [45]. In this context, our study provides novel data, as no previous research has evaluated real-world outcomes in cisplatin-treated MPE patients.

A key strength is the use of a large, contemporary cohort with treatment data collected up to 2022, ensuring the representation of patients who may benefit from recent therapeutic advances, such as targeted agents and immunotherapy, which are reshaping long-term survival prospects.

Another distinctive aspect is the integration of a standardized, multidisciplinary treatment approach. This collaborative model enhanced effectiveness and safety—particularly with high-risk drugs like cisplatin [21]—while also reducing biases and enabling a more robust and reliable analysis [46,47].

Our study has notable limitations. Firstly, it is a retrospective, single-centre cohort with a limited number of patients, introducing inherent biases. Comparable studies by Lafaras et al. (56 patients) [34], Tomkowski et al. (46 patients) [23], Maisch et al. (42 patients) [22] using cisplatin, and Shepherd et al. (58 patients) [36] using tetracyclines also faced size constraints. Furthermore, the selection of patients with ECOG PS 0–2 may lead to an underestimation of potential risks in individuals with a poorer functional status, thereby limiting the applicability of our findings to this subgroup.

Secondly, establishing a reliable causal relationship between pericardial drainage or IPIC and AEs is challenging. Electronic clinical records are collected for routine medical practice, lacking pharmacovigilance purposes, and omitting safety-influencing factors such as genetic elements. The absence of a control group is a limitation regarding the causality and applicability of the results, and comparisons with non-intervention outcomes or other effective procedures for MPE recurrences risk reduction, like prolonged catheter drainage or surgical decompression [16].

Thirdly, the variability between individuals in the number of days over which IPIC was administered might be a potential source of bias, particularly considering that one of the two recurrences occurred in a patient who only received a single dose.

Finally, although the study was designed to include different solid tumours, the predominance of lung cancer cases, mainly NSCLC, indicates that the results are primarily generalizable to this population, rather than to all tumour types. Moreover, the findings cannot be extrapolated to hematologic malignancies, which were excluded from the study.

In this single-centre study, we have evaluated the characteristics and outcomes of patients with MPE who received IPIC and, according to our experience, IPIC seems to be safe and effective. Overall, our study highlights the potential generalizability of the findings and emphasizes the importance of implementing standardized protocols and multidisciplinary teams in other hospitals to optimize patient care and treatment outcomes.

## 5. Conclusions

This study supports the effectiveness and safety of IPIC for the treatment and risk reduction of MPE recurrences. The low recurrence rate and the absence of further necessary pericardial drainage procedures may contribute towards an improved quality of life and facilitate access to systemic therapies. Further studies, preferably prospective and/or randomized studies, as well as meta-analyses, are needed to precisely assess the impact of IPIC on the survival of patients with cancer, compared to other approaches.

## Figures and Tables

**Figure 1 curroncol-32-00568-f001:**
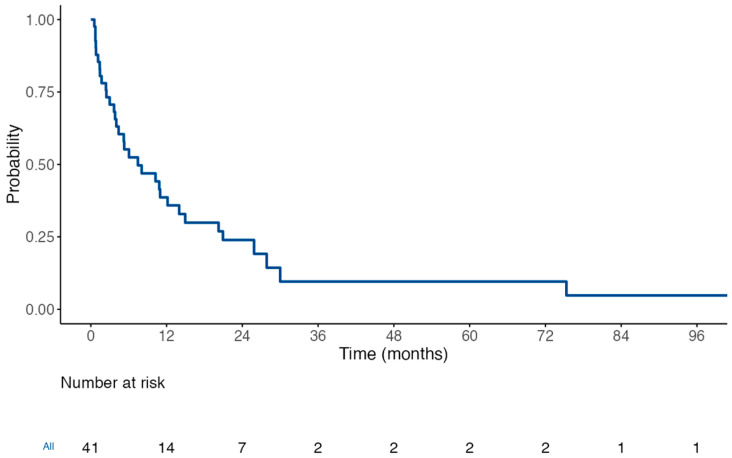
Cumulative survival of all patients since MPE diagnosis, in months. Estimated mean: 18.8 months (95% confidence interval CI 7.7–29.9 months). Estimated median: 7.5 months (95% CI 4–14.9).

**Figure 2 curroncol-32-00568-f002:**
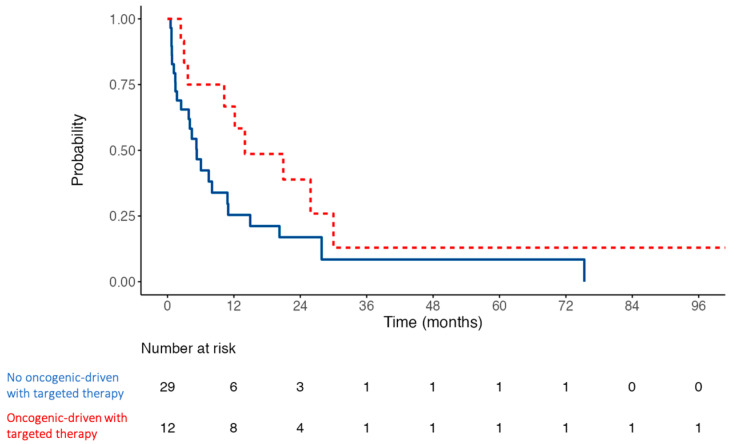
Kaplan–Meier curves of cumulative survival of patients with oncogenic-driven NSCLC treated with targeted therapies (EGFR, ALK, BRAF) compared to the rest of the patients since MPE diagnosis, in months. Estimated hazard ratio is 0.49 (95% confidence interval CI 0.22–1.07), *p*-value: 0.073.

**Table 1 curroncol-32-00568-t001:** Main characteristics of the patients included. Categorical variables are expressed in absolute numbers (percentage), and quantitative variables are expressed in medians (interquartile range).

Baseline Characteristics (n = 41)	
Sex, women	21 (51.2%)
Age	61 (51–69)
Current or former smoker	26 (63.4%)
Alcohol consumption	5 (12.2%)
Comorbidities	
Arterial hypertension	18 (43.9%)
Dyslipidaemia	8 (19.5%)
Type-2 diabetes mellitus	6 (14.6%)
Chronic heart disease	5 (12.2%)
Chronic lung disease	4 (9.8%)
Chronic kidney disease	2 (4.9%)
Chronic neurological disease	2 (4.9%)
Primary neoplasm	
Non-small cell lung cancer	32 (78%)
Small cell lung cancer	2 (5.8%)
Breast cancer	4 (9.8%)
Gastric cancer	1 (2.4%)
Malignant melanoma	1 (2.4%)
Cancer of unknown origin	1 (2.4%)

**Table 2 curroncol-32-00568-t002:** Malignant pericardial effusion, pericardiocentesis, and intrapericardial cisplatin instillation characteristics (n = 41). Categorical variables are expressed in absolute numbers (percentage) and quantitative variables are expressed in medians (interquartile range).

Main Symptom or Sign at Malignant Pericardial Effusion Diagnosis	
Dyspnoea	30 (73.1%)
Hypotension without symptoms	4 (9.8%)
Asthenia	3 (7.3%)
Chest pain or discomfort	2 (4.9%)
Syncope	1 (2.4%)
Constitutional syndrome	1 (2.4%)
Pericardiocentesis characteristics	
Initial amount of pericardial fluid drained (mL) ^1^	886 (560–1000)
Duration of pericardial drainage (days)	11 (2–25)
Adverse events related to pericardiocentesis: Transient atrial fibrillation grade 1 Chest pain grade 1 Accidental catheter withdrawal	7 (17.1%) 5 (12.2%) 1 (2.4%) 1 (2.4%)
Data regarding intrapericardial instillation of cisplatin	
Days of cisplatin instillation	4 (3–5)
Total of adverse events related to intrapericardial instillation of cisplatin	10 (24.4%)
Immediate adverse events (5 first days, those in which instillations are performed): Transient atrial fibrillation grade 1 Chest pain grade 1 Sinus tachycardia grade 1 Nausea grade 1	7 (17.1%) 3 (7.3%) 3 (7.3%) 1 (2.4%) 1 (2.4%)
Early adverse events (from sixth day to fourteen days since the first instillation): Transient atrial fibrillation grade 1 Chest pain grade 1	3 (7.3%) 2 (4.9%) 1 (2.4%)
Late adverse events (two weeks from the first instillation)	0 (0%)

^1^ At the time pericardiocentesis is performed.

## Data Availability

The original contributions presented in this study are included in the article. Further inquiries can be directed to the corresponding authors.

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
