# Peer review of "Retrospective Cohort Study of Intrapericardial Cisplatin for Risk Reduction of Malignant Pericardial Effusion Recurrence"

_curroncol, 2025, doi:10.3390/curroncol32100568_

Round 1
Reviewer 1 Report
Comments and Suggestions for Authors
Subxiphoid pericardiocentesis and intrapericardial administration of cisplatin is well established from small retrospective and multicenter retrospective studies held in the last 25 years. This retrospective cohort study from Munoz-Carrillo et al. is well organized and confirms the results of previous studies from the last 25 years. This does not offer additional information regarding the confrontation of oncologic patients suffering from malignant cardiac tamponade. Guidelines for the management of malignant cardiac tamponade include pericardiocentesis as a class I indication and intrapericardial treatment with cisplatin, especially in lung cancer patients (class IIa, level B) (2015 ESC Guidelines for the diagnosis and management of pericardial diseases: The Task Force for the Diagnosis and Management of Pericardial Diseases of the European Society of Cardiology Endorsed by: The European Association for Cardio-Thoracic Surgery).
Author Response
Dear reviewer,
We would like to sincerely thank you for your careful reading of our manuscript and for the insightful comments and suggestions you have provided. We truly appreciate the time and effort you dedicated to improving the quality of our work.
We have carefully considered each of your observations and have revised the manuscript accordingly. Below, we provide a point-by-point response to your comments, indicating the changes made in the text providing further clarification if necessary.
We hope that our revisions adequately address your concerns and improve the clarity and scientific value of our manuscript.
In summary, in your observations you mainly highlight that our study confirms the results of previous studies and lacks novelty, since intrapericardial treatment with cisplatin is already included in guidelines.
We have revised the Introduction (paragraph 3 of said section) and Discussion (paragraph 2 of said section) to clarify that while IPIC is supported in prior studies and guidelines, our contribution lines in: real-world validation, a large contemporary cohort (2009-2022) and the proposal of a standardized and multidisciplinary protocol.
Additionally, we now explicitly cite the ESC 2015 Guidelines in the Discussion (paragraph 2 of said section) to show alignment with recommended practice.
Reviewer 2 Report
Comments and Suggestions for Authors
This retrospective, single-center cohort study evaluates intrapericardial cisplatin (IPIC) in 41 patients with malignant pericardial effusion. The study is clinically relevant and provides one of the most extensive modern real-world datasets, with very low recurrence rates and manageable adverse events. The standardized multidisciplinary protocol is a strength.
However, the retrospective design and the lack of a comparator group limit the strength of the conclusions, although the outcomes are encouraging.
The subgroup survival comparison by oncogenic driver status is not related to IPIC and should either be removed or clearly reframed as descriptive background information. The manuscript should focus on recurrence prevention (which is the real endpoint) and only mention survival briefly, if at all.
In the methods, it is stated that the pericardial effusion recurrence was verified at 30 days. It is also noted that the median follow-up was 5.4 months. There were only two recurrences – it is not clear if these two recurrences were at 30 days or during the overall follow-up. This is said because previous studies have shown that recurrences can occur later than 30 days.
The discussion could benefit from a more systematic comparison with prior IPIC studies (e.g., Maisch 2002, Tomkowski 2004, Bischiniotis 2005) in terms of dose regimen, recurrence rate, and AE profile. This would situate your findings more clearly in the field.
Author Response
Dear reviewer,
We would like to sincerely thank you for your careful reading of our manuscript and for the insightful comments and suggestions you have provided. We truly appreciate the time and effort you dedicated to improving the quality of our work.
We have carefully considered each of your observations and have revised the manuscript accordingly. Below, we provide a point-by-point response to your comments, indicating the changes made in the text providing further clarification if necessary.
We hope that our revisions adequately address your concerns and improve the clarity and scientific value of our manuscript.
Comment 1: the retrospective design and the lack of a comparator group limit the strength of the conclusions, although the outcomes are encouraging.
Response: We agree that the absence of a control group limits the ability to draw causal conclusions. This is now acknowledged as a limitation in the Discussion (paragraph 17 of said section).
Comment 2: The subgroup survival comparison by oncogenic driver status is not related to IPIC.
Response: We have revised the Results (section 3.5) and rewritten the Discussion entirely to frame survival findings as descriptive only, without implying a causal relationship.
Comment 3: it is stated that the pericardial effusion recurrence was verified at 30 days. It is also noted that the median follow-up was 5.4 months. There were only two recurrences – it is not clear if these two recurrences were at 30 days or during the overall follow-up. This is said because previous studies have shown that recurrences can occur later than 30 days.
Response: We have clarified in the Results (section 3.5) that both MPE recurrences occurred within 30 days of IPIC administration.
Comment 4: The discussion could benefit from a more systematic comparison with prior IPIC studies (e.g., Maisch 2002, Tomkowski 2004, Bischiniotis 2005) in terms of dose regimen, recurrence rate, and AE profile. This would situate your findings more clearly in the field.
Response: A sentence was added in the Discussion (paragraph 7 of said section) noting that our recurrence and AE rates are consistent with previous reports (e.g., Maisch 2002, Tomkowski 2004).
Reviewer 3 Report
Comments and Suggestions for Authors
The manuscript presents valuable real-world data on intrapericardial cisplatin (IPIC) for the prevention of malignant pericardial effusion (MPE) recurrence, but several issues merit critical attention.
First, the lack of a control group is a significant limitation, as it prevents definitive conclusions on comparative effectiveness versus other established interventions such as prolonged drainage or surgical pericardial window.
While retrospective observational studies are common in rare clinical settings, the absence of matching or adjusted comparisons, even via historical cohorts or propensity scores, leaves efficacy assertions on uncertain ground.
The authors present a very low recurrence rate (4.9%), but without proper comparator arms, this could reflect patient selection rather than intervention superiority. Moreover, the study generalizes conclusions across all solid tumors, yet 78% of the cohort had NSCLC—raising questions about generalizability to other cancers like breast or gastrointestinal origins.
Another concern lies in the lack of granularity around treatment adherence and protocol deviations; only 1 dose was delivered in at least one recurrence case, yet this is not statistically integrated into the efficacy analysis.
Additionally, while adverse events are described and appear mild, the categorization lacks standardized grading (e.g., CTCAE criteria), and causality attribution is weak due to retrospective data limitations. The authors commendably attempt subgroup analysis based on oncogenic drivers, but the survival analysis (p = 0.07) fails to reach statistical significance and is over-interpreted in the discussion.
The conclusion that IPIC “enables systemic therapy and extends survival” is speculative without robust evidence linking the procedure to improved oncologic outcomes, especially given the 5.4-month median survival and lack of comparison to MPE patients not receiving IPIC. Finally, while the manuscript cites prior literature extensively, a formal meta-analysis or systematic review of IPIC versus other agents would have contextualized these findings more rigorously.
Author Response
Dear reviewer,
We would like to sincerely thank you for your careful reading of our manuscript and for the insightful comments and suggestions you have provided. We truly appreciate the time and effort you dedicated to improving the quality of our work.
We have carefully considered each of your observations and have revised the manuscript accordingly. Below, we provide a point-by-point response to your comments, indicating the changes made in the text providing further clarification if necessary.
We hope that our revisions adequately address your concerns and improve the clarity and scientific value of our manuscript.
Comment 1: the lack of a control group is a significant limitation, as it prevents definitive conclusions on comparative effectiveness versus other established interventions such as prolonged drainage or surgical pericardial window.
Response 1: We agree that the absence of a control group limits the ability to draw causal conclusions. This is now acknowledged as a limitation in the Discussion (paragraph 17 of said section).
Comment 2: the study generalizes conclusions across all solid tumors, yet 78% of the cohort had NSCLC—raising questions about generalizability to other cancers like breast or gastrointestinal origins
Response: We revised the Abstract, Simple Summary, and Discussion (paragraph 19 of said section) to clarify that our findings apply primarily to NSCLC.
Comment 3: lack of granularity around treatment adherence and protocol deviations; only 1 dose was delivered in at least one recurrence case, yet this is not statistically integrated into the efficacy analysis.
Response: We agree with your observation. We understand that deviations from the previously designed protocol can occur in real-life clinical practice studies, along with the limitations this entails. Ee have tried to highlight this in a new paragraph in the Discussion (paragraph 19 of said section).
Comment 4: while adverse events are described and appear mild, the categorization lacks standardized grading (e.g., CTCAE criteria), and causality attribution is weak due to retrospective data limitations.
Response: We have applied CTCAE v5.0 criteria to all adverse events. This is now specified in the Methods (section 2.2) and reflected in Table 2.
Comment 5: The authors commendably attempt subgroup analysis based on oncogenic drivers, but the survival analysis (p = 0.07) fails to reach statistical significance and is over-interpreted in the discussion.
Response: We rewrote the Results (section 3.5, paragraph 3) and Discussion entirely to describe this as a non-significant trend, with no suggestion of statistical significance.
Comment 6: The conclusion that IPIC “enables systemic therapy and extends survival” is speculative without robust evidence linking the procedure to improved oncologic outcomes, especially given the 5.4-month median survival and lack of comparison to MPE patients not receiving IPIC.
Response: we have rephrased this conclusion to make it mor appropriate as “The low recurrence rate and the absence of further necessary pericardial drainage procedures may contribute to improve quality of life and facilitate access to systemic therapies”.
Reviewer 4 Report
Comments and Suggestions for Authors
In this manuscript the authors present the results of a retrospective cohort study of 41 cases on the use of intrapericardial cisplatin for malignant pericardial effusion.
Line 37 “…IPIC…”
The abbreviation should be fully spelled out at its first appearance (Intra-Pericardial Instillation of Cisplatin).
Lines 50-51 “Ensuring quality and safety of intrapericardial cisplatin instillation requires a multidisciplinary approach for standardizing procedures.” Lines 304-306 “Our study reveals that standardizing treatment through a multidisciplinary team improves effectiveness and minimizes AEs, especially with high-risk drugs like cisplatin[21].” Lines 332-334 “Overall, our study highlights the potential generalizability of the findings and emphasises the importance of implementing standardized protocols and multi-disciplinary teams in other hospitals to optimize patient care and treatment outcomes” and Lines 340-341 “These findings underscore the importance of standardized procedures and a comprehensive multidisciplinary approach to optimize outcomes.”
These repetitious claims are superfluous and unjustified. It is common knowledge that a standardized protocol facilitates outcome data analysis. Management of malignant intrapericardial effusion inevitably requires a multidisciplinary approach. In this retrospective review the authors present no data to show how their “standardized procedures” and “comprehensive multidisciplinary approach” actually optimized the outcomes.
Page 2 insert:
“Intrapericardial cisplatin” does not match with its abbreviation IPIC.
In the pie chart, the blue slices should be labeled.
Line 62 “…haematological malignancies (~15%)…” and Line 92 “diagnosed with a solid neoplasm and MPE treated with IPIC”
The second statement means cases of hematological malignancies were excluded in this study. The lack of hematological malignancy cases in this series should be noted in Simple Summary, Abstract and Discussion.
Lines 220-224 “In a subgroup analysis of patients with targetable oncogenic alterations (EGFR, ALK, BRAF), the median overall survival was longer compared to the overall cohort. Kaplan-Meier curves for these subgroups are shown in Figure 2, indicating a median survival of 14 months (95% confidence interval, 10.2 - not reached; p-value of 0.07).”
There are three problems. Firstly, conventionally a P-value of 0.07 is not considered statistically significant. Secondly, the text (…compared to the overall cohort.) is inconsistent with the legend of Figure 2 (…compared to the rest of the patients….). Thirdly, NSCLC with targetable oncogenic alterations should only be compared to NSCLC without such alterations.
Line 280-281: “In our series, AEs related to pericardiocentesis or drug instillation occurred in less than 20% of patients”
This statement contradicts the data of 24.4% presented in the Results section, page 2 insert and Table 2.
Lines 301-302 “……Consequently, cytotoxic agents such as cisplatin offer a more interesting profile compared to other drugs, such as sclerosing agents”
Since there are no head-on comparative studies, the verb “offer” should be reviised to “may offer”.
Table 2 “Echocardiographic pericardial effusion volume estimation”
The data presented in this section is meaningless. A classification of the volumes drained in the section “Pericardial characteristics” would be more informative.
Comments on the Quality of English Language
English should be improved for clarity.
Author Response
Dear reviewer,
We would like to sincerely thank you for your careful reading of our manuscript and for the insightful comments and suggestions you have provided. We truly appreciate the time and effort you dedicated to improving the quality of our work.
We have carefully considered each of your observations and have revised the manuscript accordingly. Below, we provide a point-by-point response to your comments, indicating the changes made in the text providing further clarification if necessary.
We hope that our revisions adequately address your concerns and improve the clarity and scientific value of our manuscript.
Comment 1: The abbreviation should be fully spelled out at its first appearance (Intra-Pericardial Instillation of Cisplatin).
Response: corrected.
Comment 2: repetitious claims regarding standardized protocol and multidisciplinary.
Response: We apologize for the redundancy in the original text. We've tried to rewrite the sections where this occurred to make it more concise and accurate regarding its relevance to the work, especially deleting repeated mentions in the simple summary, abstract and conclusions.
Comment 3: In the pie chart, the blue slices should be labelled.
Response: labels have been added.
Comment 4: The lack of haematological malignancy cases in this series should be noted in Simple Summary, Abstract and Discussion.
Response: corrected.
Comment 5: Survival comparison data unclear in the subgroup analysis of patients with targetable oncogenic alterations.
Response: We revised and rewrote the Results and Figure 2 caption to clarify the previous inconsistence and highlight that the comparison does not reach statistically significance.
Additionally, we have also compared NSCLC with targetable oncogenic alterations to NSCLC without such alterations as suggested, but does not reach statistically significance either (p=0.18). We have added this information to the Results (section 3.5, paragraph 3).
Comment 6: AE rates were inconsistent, stated as <20% vs. 24.4%.
Response: All AE rates have been revised for consistency (24.4%).
Comment 7: speculative language in the sentence “offers better profile”.
Response: This was changed to “may offer” in the Discussion (paragraph 12 of said section).
Comment 8: The data presented in table 2 is meaningless. A classification of the volumes drained in the section “Pericardial characteristics” would be more informative.
Response: We replaced the categorical classification with the median volume drained (in mL) in Table 2.
Reviewer 5 Report
Comments and Suggestions for Authors
This study addresses an important clinical problem, as malignant pericardial effusion (MPE) is a life-threatening oncologic emergency with frequent recurrence. By implementing a standardized multidisciplinary protocol for intrapericardial cisplatin (IPIC) instillation, the authors provide a reproducible model with practical therapeutic implications. The inclusion of 41 patients over 13 years makes this one of the larger real-world IPIC cohorts, and the outcomes—low recurrence (4.9%), manageable adverse events (24%), and a median survival of 5.4 months—are clearly presented, including subgroup analyses for oncogenic drivers.
However, the retrospective, single-centre design without a control group limits generalizability, and selection of ECOG 0–2 patients may underestimate risks in frailer populations. While the low recurrence rate is encouraging, survival benefits cannot be directly attributed to IPIC given the impact of systemic therapy, and adverse event reporting would be strengthened by CTCAE grading. The predominance of NSCLC (78%) also restricts applicability to other tumor types. To improve the manuscript, the authors should contextualize findings against historical comparators, expand adverse event grading, address long-term safety, provide more detail on subgroups and patients receiving fewer instillations, and streamline repetitive discussion points.
Author Response
Dear reviewer,
We would like to sincerely thank you for your careful reading of our manuscript and for the insightful comments and suggestions you have provided. We truly appreciate the time and effort you dedicated to improving the quality of our work.
We have carefully considered each of your observations and have revised the manuscript accordingly. Below, we provide a point-by-point response to your comments, indicating the changes made in the text providing further clarification if necessary.
We hope that our revisions adequately address your concerns and improve the clarity and scientific value of our manuscript.
Comment 1: the retrospective, single-centre design without a control group limits generalizability.
Response: We agree with your observation. We have highlighted it in the Discussion as a limitation of our study.
Comment 2: selection of ECOG 0–2 patients may underestimate risks in frailer populations.
Response: This is now acknowledged as a limitation in the Discussion (paragraph 16 in said section).
Comment 3: survival benefits cannot be directly attributed to IPIC given the impact of systemic therapy.
Response: we have rewritten the Discussion to focus on malignant pericardial effusion recurrence and not in survival benefits since, as you noted, can not be attributed directly to IPIC.
Comment 4: adverse event reporting would be strengthened by CTCAE grading.
Response: CTCAE grading has been added to adverse event reporting.
Comment 5: The predominance of NSCLC (78%) also restricts applicability to other tumour types.
Response: we have highlighted this fact and the limitation it implies in different sections of the article and has been explicitly reflected in the Discussion (paragraph 20 of said section).
Comment 6: general observations and streamline repetitive discussion points.
Response: The Discussion has been entirely rewritten and tightened by removing repeated points and consolidating key findings.
Round 2
Reviewer 2 Report
Comments and Suggestions for Authors
The manuscript is improved, and now it can be published.
Reviewer 3 Report
Comments and Suggestions for Authors
They covered all my points.